

# Meiofaunal communities and nematode diversity characterizing the Secca delle Fumose shallow vent area (Gulf of Naples, Italy)

Elisa Baldrighi[1,2], Daniela Zeppilli[2], Luca Appolloni[3,4],
Luigia Donnarumma[3,4], Elena Chianese[3], Giovanni Fulvio Russo[3,4] and
Roberto Sandulli[3,4]

[1] Istituto per le Risorse Biologiche e le Biotecnologie Marine (IRBIM), Consiglio Nazionale delle Ricerche (CNR), Lesina, Italy
[2] Laboratoire Environnement Profond, Institut Français de Recherche pour l'Exploitation de la MER (IFREMER), Plouzané, France
[3] Department of Science and Technology (DiST), Parthenope University of Naples, Naples, Italy
[4] Consorzio Nazionale Interuniversitario per le Scienze del Mare (CoNISMa), Rome, Italy

Corresponding author
Elisa Baldrighi,
elisa.baldrighi@irbim.cnr.it

## ABSTRACT

Hydrothermal venting is rather prevalent in many marine areas around the world, and marine shallow vents are relatively abundant in the Mediterranean Sea, especially around Italy. However, investigations focusing on the characterization of meiofaunal organisms inhabiting shallow vent sediments are still scant compared to that on macrofauna. In the present study, we investigated the meiobenthic assemblages and nematode diversity inhabiting the Secca delle Fumose (SdF), a shallow water vent area located in the Gulf of Naples (Italy). In this area, characterized by a rapid change in the environmental conditions on a relative small spatial scale (i.e., 100 m), we selected four sampling sites: one diffusive emission site (H); one geyser site (G) and two inactive sites (CN, CS). Total meiofauna abundance did not vary significantly between active and inactive sites and between surface and deeper sediment layers due to a high inter-replicate variability, suggesting a pronounced spatial-scale patchiness in distribution of meiofauna. Nematofauna at site H presented the typical features of deep-sea vents with low structural and functional diversity, high biomass and dominance of few genera (i.e., *Oncholaimus*; *Daptonema*) while from site G we reported diversity values comparable to that of the inactive sites. We hypothesized that site G presented a condition of "intermediate disturbance" that could maintain a high nematode diversity. Environmental features such as sediment temperature, pH, total organic carbon and interstitial waters ions were found to be key factors influencing patterns of meiofauna and nematofauna assemblages at SdF. Even though the general theory is that nematodes inhabiting shallow vent areas include a subset of species that live in background sediments, this was not the case for SdF vent area. Due to a marked change in nematode composition between all sites and to the presence of many exclusive species, every single investigated site was characterized by a distinct nematofauna reflecting the high spatial heterogeneity of SdF.

## INTRODUCTION

Scientific explorations have demonstrated the importance of hydrothermally influenced habitats both in the deep-sea and in shallow coastal habitat (*Melwani & Kim, 2008*). Shallow vents are often associated with active plate boundaries, and consequently volcanic or seismic activities are related to sites of littoral and sub-littoral thermal venting (*Tarasov et al., 1999*). The environmental conditions in shallow-water vents strongly differ from the surrounding seafloor in terms of both increased temperature and enrichment in reduced chemical compounds such as sulfide, methane, manganese, iron and arsenic (*Prol-Ledesma et al., 2004*). Fluids formation commonly take place from relatively deep sources (1–2 km depth) and these natural fluid emissions may be able to alter sea-water geochemistry (*Di Bella et al., 2016*). Pore-water temperatures tend to significantly increase compared to ambient conditions (*Pichler, Veizer & Hall, 1999*). Furthermore, numerous studies have reported vent fluids with low salinity and acidic pH (*Melwani & Kim, 2008* and literature therein). The high temperature, coupled with gas release and variable chemical conditions due to hydrothermal activity may create biologically stressful environments.

Differently from deep-sea hydrothermal vents based exclusively on chemosynthetic primary production, shallow-water vents are characterized by the presence of light that coupled with that of geothermal fluids, promotes both photo- and chemosynthetic primary production (*Sorokin, Sorokin & Zakuskina, 1998*; *Tarasov et al., 2005*). Most ecological studies investigating shallow water hydrothermal vents were focused on microbial communities, which are strongly influenced by the hydrothermal activity both within the water column and on the seabed (*Judd & Hovland, 2007*; *Di Bella et al., 2016*). Shallow-water vents can differ in terms of faunal density, diversity and dominance (i.e., metazoan organisms) from the surroundings and from each other, depending on the degree and effects of the venting activity (*Jones, 1993*; *Panieri et al., 2005*, *2006a*; *Melwani & Kim, 2008*; *Wildish et al., 2008*).

At deep-sea hydrothermal vents, an increase in number of meiobenthic animals coupled with a reduction in the diversity compared to non-vent areas has been reported (*Copley et al., 2007* and literature therein; *Zeppilli et al., 2018*). Nematodes and copepods are the most abundant taxa and studies conducted along the East Pacific Rise reported that usually copepods are the initial dominant meiofaunal colonizers of "new" vent mussel beds, with a general increase in the percentage ratio of nematodes to copepods over time (*Flint et al., 2006*; *Copley et al., 2007*). Deep-water hydrothermal vents in general do not show high nematode densities or biomass and also their diversity is much lower than that in surrounding deep-sea sediments (*Vanreusel et al., 2010*). Despite their ubiquitous distribution in tectonically active coastal zones, shallow-water vents have been less explored than deep-sea vents in terms of biodiversity and adaptations to extreme conditions (*Colangelo et al., 2001*; *Tarasov et al., 2005*). Investigations focusing on the characterization of meiofaunal organisms inhabiting shallow vent sediments are still scant compared to that on macrobenthic communities and they span different geographical areas, from Mediterranean Aegean Sea (*Dando et al., 1995*; *Fitzsimons et al., 1997*;

*Thiermann, Windoffer & Giere, 1994*; *Thiermann et al., 1997*), Tyrrhenian Sea (*Colangelo et al., 2001*; *Panieri et al., 2005*, *2006a*; *Di Bella et al., 2016*), Adriatic Sea (*Panieri, 2006b*), Strait of Sicily (*Sandulli et al., 2015*) to New Zealand (*Kamenev et al., 1993*), Papua New Guinea (*Tarasov et al., 1999*), Indonesian archipelago (*Zeppilli & Danovaro, 2009*), the shallow sub-polar region of the Mid Atlantic Ridge (*Fricke et al., 1989*), Azores (*Cardigos et al., 2005*), and including the comprehensive reviews by *Tarasov et al. (2005)* and *Zeppilli et al. (2018).*

Shallow vents are characterized by non-endemic meiofauna that show higher diversity and abundance than in background sediments (*Tarasov et al., 2005*). While deep-sea vents are inhabited by unique nematode assemblages, at least at species level (*Vanreusel, Van Den Bossche & Thiermann, 1997*; *Thiermann et al., 1997*), shallow-water vent nematodes include a subset of species that can both live in the background sediments and cope with extreme conditions (*Zeppilli & Danovaro, 2009*). Also for copepods, shallow vent assemblages seem to be the result of colonization from adjacent areas (*Colangelo et al., 2001*; *Zeppilli & Danovaro, 2009*).

Hydrothermal venting in shallow water is a common phenomenon and marine shallow vents are abundant in the Mediterranean, in particular in the Tyrrhenian Sea (*Hall-Spencer et al., 2008*). The present work represents the first investigation on the meiobenthic communities inhabiting Secca delle Fumose (SdF) shallow water vent area located in the Underwater Archaeological Park of Baia (Gulf of Naples, Italy).

The peculiarity of this active area is the rapid change in the extreme environmental conditions, from the diffusive vent site and geyser site with hot temperatures to inactive sites (see below), at a relatively small spatial scale that is, 100 m. SdF can be considered an example of how hydrothermal flux can vary at small scale in the vent habitat (*Flint et al., 2006*), creating a gradient in the environmental conditions, from extreme to ambient conditions.

The present article reports a first insight into the meiofaunal and nematode communities of SdF shallow vent area and describes the distribution and diversity of meiobenthic organisms in relation to seawater chemistry and sediment characteristics. Due to changes in stress regime and environmental conditions, we expect different meiofauna and nematofauna assemblages inhabiting the distinct sampling sites.

## MATERIALS AND METHODS

### Site description

The study area of SdF is located in the north-west side of the Gulf of Naples (Baia Underwater Park MPA, Tyrrhenian Sea, Italy; Fig. 1). The coastline consists mainly of beaches and tuff cliffs. Volcanic activity is still evident both for bradyseism process, which led to the immersion of many Roman structures, and for sulfurous emanations. The interaction between natural processes and human activities produced a natural environment characterized by an extreme habitats' variety (Fig. 1). Photophilous algae habitat (AP) is observed on docks, while strong interdigitation between photophilous populations, which are setting up on the parts most exposed to light and sciaphilous populations (C) setting in the cavities (AP-C), is present on Roman artifacts. Superficial
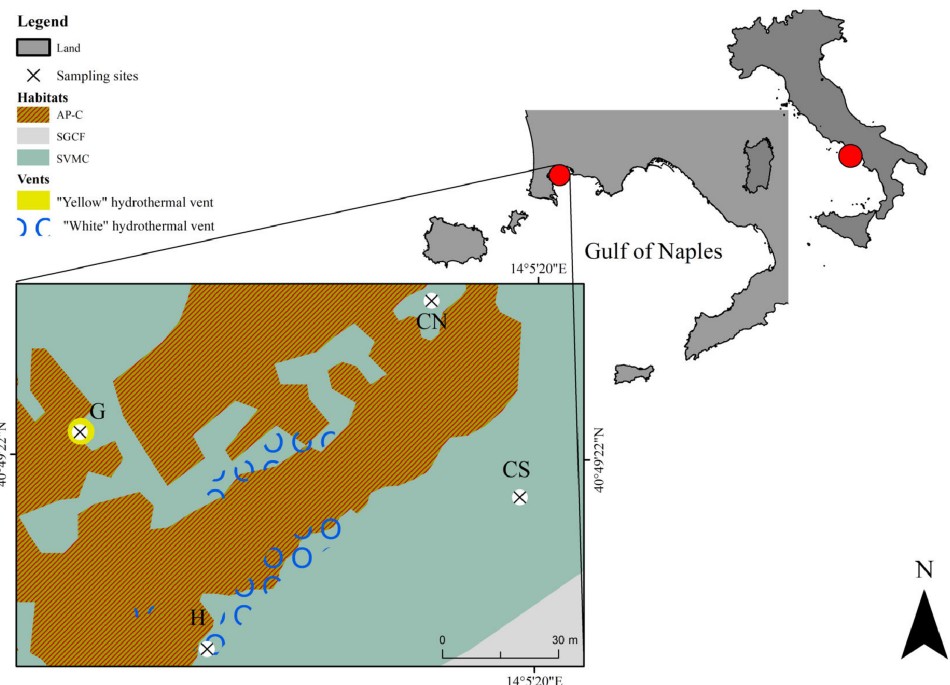

**Figure 1 Map showing the location of the sampling area Secca delle Fumose in underwater park of Baia MPA located in the Gulf of Naples (Tyrrhenian Sea, Italy).** Selected sampling sites are: the diffusive emission site (H) called "White" for the presence of white microbial mats; the geyser site (G) called "Yellow" for the sulfur deposits; the inactive sites north (CN) and south (CS). The different habitats are indicated as follow: AP-C, photophilous algae and sciaphilous cavities; SGCF, coarse sands and fine gravels; SVMC, superficial muddy sands. The base map was performed using a Side Scan Sonar (SSS) Klein 3900—450 kHz. The SSS used 2 transducers to transmit a high-frequency acoustic signal towards the sea floor and the return signal was picked up by the SSS and transmitted in real time to a graphic recorder on board ship. The ship's position was continuously recorded from a positioning satellite system (DGPS and Gyro GNSS Hemisphere Vector gnss VS 330). SSS and navigation data were processed using SonarPro®.

muddy sands in sheltered water habitat (SVMC) dominates soft bottoms. Near the submerged structures SVMC habitats are enriched with fragments come from the disintegration of the artifacts and from carbonate exoskeletons of benthic organisms; here are recorded species belonging to the coarse sands and fine gravels under the influence of bottom currents habitat (SGCF) (as previously described in *Appolloni et al. (2018)*). SdF is a submarine relief consisting of a network of ancient Roman pillars, among which thermal vents releasing hot gas-rich hydrothermal fluids occur (*D'Auria et al., 2011*). This area has been previously investigated from a geological point of view (*Todesco, 2009*; *D'Auria et al., 2011* and literature therein), and, very recently, with the only macrobenthic and environmental published study so far known (*Donnarumma et al., 2019*). All details regarding the sampling site are reported in *Donnarumma et al. (2019)*.

## Sampling methods and samples processing

For our study, we selected four sampling sites, at 9–14 m water depth, over a total of ~7,500 m$^2$ sampling area: one diffusive emission site (H, "White") characterized by the presence of white microbial mats covering the soft bottom; one geyser site (G, "Yellow") at

65 m distance from the H site, with surrounding rocky substrate covered by yellow sulfur deposits and with hot water emissions reaching 80 °C at the sediment surface; two inactive sites (CN and CS) located at a distance of 100 m from the active sites H and G (Fig. 1). Environmental data were collected as previously describe in *Donnarumma et al. (2019)*. Interstitial waters to assess ions and heavy metals concentrations and sediment samples (three random replicates) were collected at the four selected sites. Cylindrical corers (internal diameter 5.5 cm) were used to collect sediment for grain size, total organic carbon (TOC) and meiofaunal communities by scuba diving. The meiofauna sediment cores were sliced into four layers 0–1, 1–3, 3–5 and 5–10 cm and fixed in buffered 4% formalin/seawater.

## Environmental variables

All details on the environmental variable analyses considered in the present study that is, interstitial waters ion ($Na^+$, $Cl^-$, $K^+$, $Mg^{2+}$, $Ca^{2+}$, $NO_3^-$, $SO_4^{2-}$ and $S^{2-}$) and heavy metal (Zn, Pb, Cd and Cu) concentrations, sediment grain size (as percentage of gravel, sand and mud) and TOC are reported in *Donnarumma et al. (2019)*.

## Meiobenthos and nematode analyses

Sediment samples were sieved through a 1,000 µm mesh, and a 30 µm mesh was used to retain the smallest organisms. The fraction remaining on the latter sieve was re-suspended and centrifuged with Ludox HS40 according to *Heip, Vincx & Vranken (1985)*.
All meiobenthic animals were counted and classified per taxon under a stereomicroscope. From each sample, approximately 100–120 nematodes were picked out, transferred to anhydrous glycerin following the formalin–ethanol–glycerin protocol (*De Grisse, 1969*), and mounted on paraffin ring glass slides for microscopic identification. Nematodes were identified at genus level and diversified in putative morphotypes as sp1, sp2, etc. According to the main original species descriptions (*Platt & Warwick, 1983*, *1988*; *Warwick, Platt & Somerfield, 1998*; *Tchesunov & Schmidt-Rhaesa, 2014*) and pictorial keys available on the Nemys website (*Bezzerra et al., 2018*). Nematode biomass was calculated from the biovolume using the *Andrassy (1956)* formula ($V = L \times W^2 \times 0.063 \times 10^{-5}$, in which $L$ is the body length and $W$ is the body width). Species richness (SR) was calculated as the total number of species collected at each site; the expected number of species for a theoretical sample of 51 specimens, ES(51), was selected and *Pielou's (1975)* evenness (J′) was estimated.

Furthermore, nematode genera were categorized in four feeding guilds based on their buccal cavity morphology as described by *Wieser (1953)*. Feeding guilds included "selective deposit feeders" (Group 1A, small buccal cavity without teeth), "non-selective deposit feeders" (Group 1B, large buccal cavity without teeth), "epistrate feeders" (Group 2A, small buccal cavity with teeth) and "predators/scavengers" (Group 2B, lager buccal cavity with teeth). The index of trophic diversity (ITD) as ITD = $g_12 + g_22 + g_32\ldots + g_n2$, where $g$ is the relative contribution of each trophic group to the total number of individuals and $n$ is the number of trophic groups (*Gambi, Vanreusel & Danovaro, 2003*). For $n = 4$, ITD ranges from 0.25 (highest trophic diversity, i.e., the 4 trophic guilds account for 25% each)

to 1.0 (lowest diversity i.e., one trophic guild accounts for 100% of nematode density). To identify the life strategy of nematodes, the maturity index (MI) was calculated according to the weighted mean of the individual genus scores: MI = $\Sigma v(i)i(i)$, where $v$ is the c–p value (colonizers–persisters) of genus $i$ as given in the appendix of *Bongers, Alkemade & Yeates (1991)* and $i(i)$ is the frequency of that genus.

## Data analysis

Uni- and multivariate analyses were carried out in order to assess differences in several descriptors of meiofauna (i.e., abundance, number of taxa, assemblage composition) and nematode assemblage composition (i.e., abundance, biomass, SR, ES(51), J', ITD and MI) between sites and layers. Faunal data were Log$_{(X+1)}$ transformed and analyzed using tests based on Bray–Curtis similarity matrices (multivariate analyses) and Euclidean similarity matrices (univariate analyses).

The sampling design included two fixed and orthogonal factors: site (4 levels: G, H, CN and CS) and layer (4 levels: 0–1, 1–3, 3–5 and 5–10 cm). The distance-based permutation analysis of variance (PERMANOVA; *Anderson, Gorley & Clarke, 2008*) in either univariate (separately for each meiofauna and nematode diversity index) or multivariate data (for both meiofaunal and nematode assemblages) was used for testing for differences in taxonomic composition between sites and layers. Pair-wise tests were carried out to verify the significance of the differences among sites and layers if any were observed in the main test.

Afterwards, the relative contribution of each nematode species to the average dissimilarities between sites and layers was calculated using SIMPER test (using 70% as cutoff) to determine the contribution of each species (*Clarke & Warwick, 2001*). The diversity in nematode community composition is expressed as percentages of dissimilarity (*Gray, 2000*). A CLUSTER analysis (group average) was carried out and a similarity profile test (SIMPROF) permutational routine was applied to test for the significance of genuine clustering on meiofaunal assemblage composition and nematode species composition characterizing the sampling sites.

Multivariate multiple regression analyses (DistLM forward, *Anderson, Gorley & Clarke, 2008*) with a forward selection of the independent variables and 4,999 permutation of residuals were performed to test the influence of abiotic variables (sediment and interstitial water variables) on meiofaunal abundance, richness of taxa and taxonomic composition, nematode biomass, species composition and diversity indices, nematode trophic diversity and nematode life strategies. All the analyses were carried out by means of the software PRIMER v6.0+ (*Clarke & Gorley, 2006*).

## RESULTS

### Environmental variables

Environmental characteristics of sampling sites at SdF are reported and described in detail in *Donnarumma et al. (2019)*. The H site, the diffusive emission site in the southern sector, was characterized by the highest sediment temperature (37.53 ± 2.28 °C) and by the lowest pH value of (7.56 ± 0.05). At this site, sediment was mainly composed by the

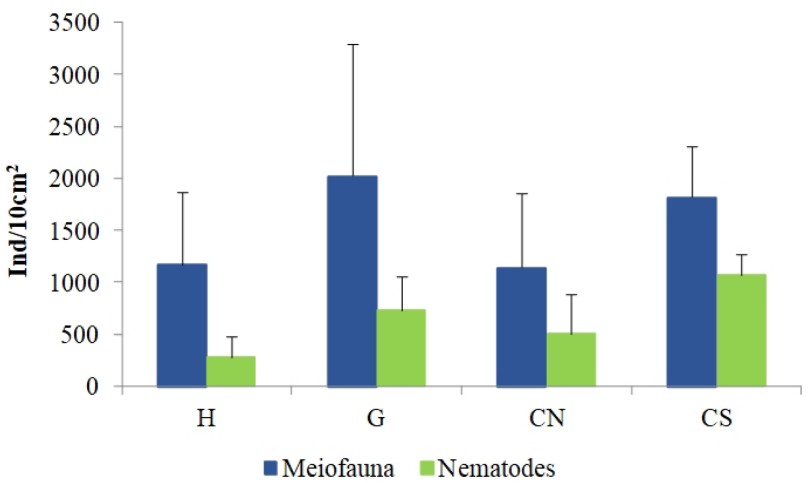

**Figure 2 Total meiofaunal and nematode abundances.** Reported are mean values (±SD) at all investigated sites: the diffusive emission site (H); the geyser site (G); the inactive sites north (CN) and south (CS).

sandy fraction enriched in TOC content and the soft bottom presented a coverage of white microbial mat. At G site (geyser, northern sector), located at 65 m from the H site, the sediment temperature reported was 29.1 ± 2.81 °C. This site was the only one characterized by the presence of sulfur ion $S^{2-}$ into the interstitial waters. CN and CS constituted the inactive sites at a distance of 100 m from G and H sites. The sediment temperature (21.8 °C) and pH (8.1) values were comparable to the surrounding environment. All sampling sites were clearly differed due to many environmental features that changed over a relatively small spatial scale (100 m) and indicating a marked spatial heterogeneity (*Donnarumma et al., 2019*).

## Meiofauna abundance, taxonomic composition and distribution

The mean meiofaunal abundances and abundance of each taxon at each sampling sites and along the sediment layers are reported in Table S1. The total meiofaunal abundance ranged from 1,142 ± 713.8 ind./10 cm$^2$ (mean ± standard error, hereafter) at CN site to 2,023 ± 1,270.2 ind./10 cm$^2$ at G site (Fig. 2). At all sampling sites, with the only exception for the inactive site CN, an increase of meiofaunal abundance from the surface sediment (0–1 cm) to the deeper sediment layer (5–10 cm; Fig. 3) was recorded. Nevertheless, PERMANOVA analysis did not detect any significant effect of factors "site" and "layer" on total meiofauna abundance (Table 1). The total nematode abundances followed the total meiofaunal trends (Figs. 2 and 3). Significant lower values in the total nematode abundance were reported at site H compared to sites G and CS (Table 1). The abundance of nematodes increased significantly from the top 1 cm to the deepest layer at sites H and CS (Fig. 3; Table 1). A total of 27 higher taxa, including Foraminifera and Ciliata, were identified from the sediments of SdF (Table S1). The total number of meiobenthic taxa ranged from 11 ± 1 at site H to 20 ± 1 at site G. PERMANOVA analysis indicated that "site" and the combined effect of S × L were the factors explaining meiofaunal diversity variability in the number of major taxa (Table 1). In detail, the pair-wise test

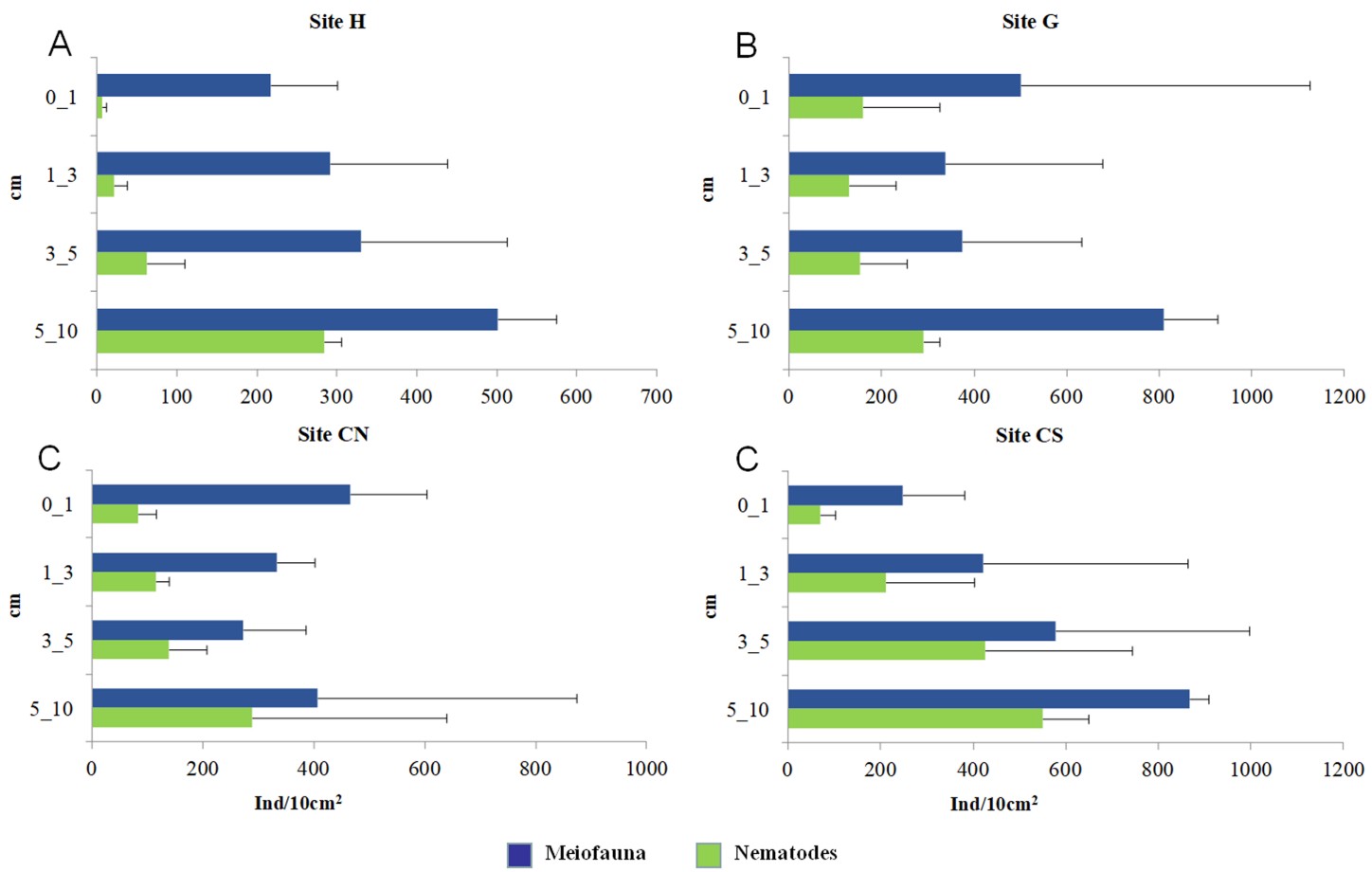

**Figure 3 Vertical distribution of total meiofaunal and nematode abundances.** Reported are mean values (±SD) at four sampling sites (A–D).

showed that site H was characterized by a significant lower meiofaunal diversity compared to all other sites (Table 1). Meiofaunal taxonomic composition changed significantly between sites (Table 1), the pair-wise test detected major differences between site H vs. all the other sites and the CLUSTER analysis clearly highlighted this separation (Fig. 4).

With the only exception of site H, Nematoda were the most represented taxon (36–57%), followed by Copepoda Harpacticoida and their *nauplii* (21–30%), Foraminifera (12–13) and Polychaeta (4–8%) (Table S1; Fig. 5). Other taxa, such as Tardigrada, Ciliata, Ostracoda, Gastropoda and Halacarida typically occurred in lower numbers (2–5%) or only occasionally (e.g., Kinorhyncha, Gastrotricha, Amphipoda, Tanaidacea, Cumacea, Isopoda). At site H, harpacticoid copepods and their *nauplii* were the most represented groups (41%) followed by Ciliata (29.5%) and Nematoda (24%). SIMPER analysis on meiofaunal taxonomic composition characterizing the four sampling sites reported a dissimilarity percentage between site H and all the other sites ranging from 27% to 31%. The dissimilarity was mainly due to a different contribution of taxa, in term of abundances, at the different sampling sites and secondly to the absence of Tardigrada and Gastrotricha at the diffusive site (H). Ostracoda, Polychaeta, Gasteropoda and Halacarida,

**Table 1 Results of PERMANOVA and PAIR-WISE tests for differences in meiofaunal abundance, number of taxa, meiofaunal taxonomic composition, nematode biomass and nematode species composition among sites (S) and layers (L).**

| A | Source | df | MS | F | P | B | Source | | P |
|---|---|---|---|---|---|---|---|---|---|
| Total meiofauna abundance | Site | 3 | 1.194 | 1.244 | ns | Meiofaunal no taxa | Site | H vs. G | 0.001 |
| | Layer | 3 | 2.243 | 2.337 | ns | | | H vs. CN | 0.001 |
| | S × L | 9 | 0.591 | 0.616 | ns | | | H vs. CS | 0.001 |
| | Residual | 29 | 0.960 | | | Meiofaunal composition | Site | H vs. G | 0.009 |
| | Total | 44 | | | | | | H vs. CN | 0.002 |
| Meiofaunal no taxa | Site | 3 | 6.079 | 14.632 | 0.001 | | | H vs. CS | 0.006 |
| | Layer | 3 | 1.049 | 2.524 | ns | Total nematode abundance | Site | H vs. G | 0.024 |
| | S × L | 9 | 1.053 | 2.535 | 0.023 | | | H vs. CS | 0.006 |
| | Residual | 29 | 0.415 | | | | Layer (within levels | 0–1 vs. 5–10 | 0.001 |
| | Total | 44 | | | | | H and CS) | 1–3 vs. 5–10 | 0.002 |
| Meiofaunal composition | Site | 3 | 69.482 | 2.5462 | 0.002 | | | | |
| | Layer | 3 | 30.437 | 1.1154 | ns | Total nematode biomass | Site | H vs. G | 0.001 |
| | S × L | 9 | 24.074 | 0.8822 | ns | | | H vs. CN | 0.005 |
| | Residual | 29 | 27.289 | | | | | H vs. CS | 0.026 |
| | Total | 44 | | | | | | G vs. CN | 0.002 |
| Total nematode abundance | Site | 3 | 2.953 | 5.067 | 0.007 | | | G vs. CS | 0.001 |
| | Layer | 3 | 4.622 | 7.931 | 0.001 | | Layer (within levels | 0–1 vs. 3–5 | 0.012 |
| | S × L | 9 | 0.556 | 0.954 | ns | | H, G and CN) | 0–1 vs. 5–10 | 0.001 |
| | Residual | 29 | 0.583 | | | | | 1–3 vs. 3–5 | 0.014 |
| | Total | 44 | | | | | | 1–3 vs. 5–10 | 0.001 |
| Total nematode biomass | Site | 3 | 12.539 | 4.180 | 0.001 | Nematode composition | Site | H vs. G | 0.001 |
| | Layer | 3 | 9.366 | 3.122 | 0.001 | | | H vs. CN | 0.001 |
| | S × L | 9 | 15.528 | 1.725 | 0.001 | | | H vs. CS | 0.001 |
| | Residual | 29 | 9.012 | 0.311 | | | Site | G vs. CN | 0.001 |
| | Total | 44 | 44.000 | | | | | G vs. CS | 0.001 |
| Nematode composition | Site | 3 | 64,495 | 21498 | 0.001 | | | CN vs. CS | 0.001 |
| | Layer | 3 | 5,725.7 | 1908.6 | ns | ES(51); SR | Site | H vs. G | 0.005 |
| | S × L | 9 | 15,263 | 1695.9 | ns | | | H vs. CN | 0.003 |
| | Residual | 29 | 49,023 | 1690.4 | | | | H vs. CS | 0.001 |
| | Total | 44 | 135,440 | | | J | Site | H vs. G | 0.001 |
| ES(51); SR | Site | 3 | 758.17 | 39.535 | 0.001 | | | H vs. CN | 0.001 |
| | Res | 8 | 19.177 | | | | | H vs. CS | 0.001 |
| | Total | 11 | | | | | | CS vs. G | 0.02 |
| J | Site | 3 | 1,025.2 | 163.61 | 0.001 | | | CS vs. CN | 0.007 |
| | Res | 8 | 6.2663 | | | ITD | Site | H vs. CS | 0.005 |
| | Total | 11 | | | | | | CS vs. G | 0.014 |
| ITD | Site | 3 | 357.23 | 14.045 | 0.002 | | | CS vs. CN | 0.007 |
| | Res | 8 | 25.435 | | | MI | Site | H vs. G | 0.002 |
| | Total | 11 | | | | | | H vs. CN | 0.001 |
| MI | Site | 3 | 36.44 | 47.043 | 0.001 | | | H vs. CS | 0.03 |
| | Res | 8 | 0.77461 | | | | | CS vs. G | 0.003 |
| | Total | 11 | | | | | | CS vs. CN | 0.004 |

**Note:**
Reported are (A) results of PERMANOVA Main tests and (B) Pair-wise tests. Df, degree of freedom; $F$, F-statistic; ns, not significant.

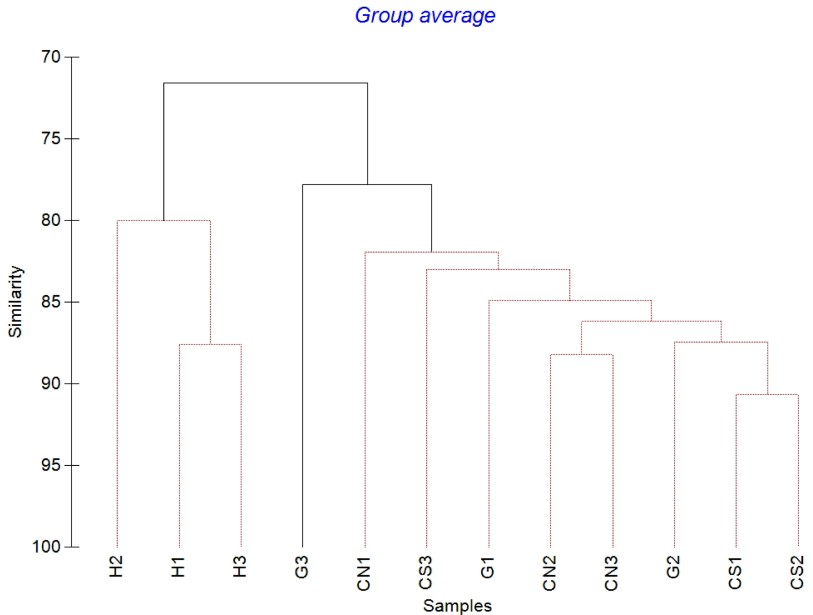

**Figure 4 Cluster analysis with SIMPROF test performed on meiofaunal community composition.** Red lines indicate statistically significant groupings according to the SIMPROF routine. All replicates are shown and indicated as: H1, H2 and H3; G1, G2 and G3; CN1, CN2 and CN3; CS1, CS2 and CS3.

**Figure 5 Meiofauna taxonomic composition.** *Others*: Bivalvia, Gasteropoda, Scaphopoda, Polyplacophora, Oligochaeta, Amphipoda, Isopoda, Cumacea, Tanaidacea, Cladocera, Halacarida, Tardigrada, Kinorhyncha, Gastrotricha, Platyhelminthes, Nemertea, Sipuncula, Porifera, Rotifera, Cnidaria and Chaetognatha.

occurred frequently (2–8%) at sites G, CN and CS; other taxa, such as Ciliata and Copepoda were found in high abundances at site H (Table S1; Fig. 5). Meiofauna taxonomic composition did not change significantly along the vertical sediment profile. Overall, nematode abundances increased with depth layer, Copepoda (with their *nauplii*), on the opposite, decreased deepening into the sediment. These patterns characterized all sampling sites (Table S1; Fig. 6).

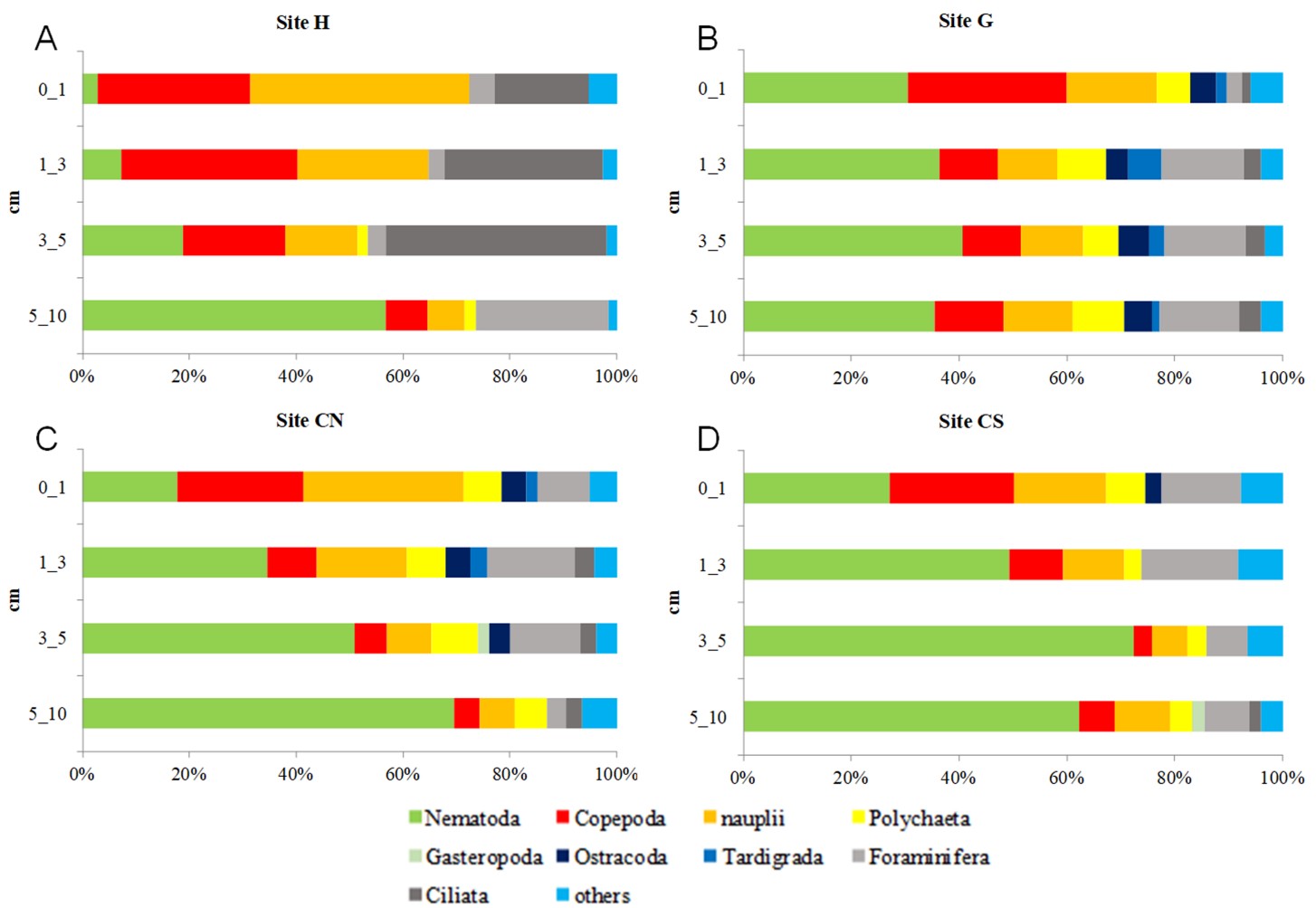

**Figure 6 Vertical meiofauna taxonomic composition four sampling sites (A–D).** *Others*: Bivalvia, Gasteropoda, Scaphopoda, Polyplacophora, Oligochaeta, Amphipoda, Isopoda, Cumacea, Tanaidacea, Cladocera, Halacarida, Tardigrada, Kinorhyncha, Gastrotricha, Platyhelminthes, Nemertea, Sipuncula, Porifera, Rotifera, Cnidaria and Chaetognatha.

## Nematode biomass, structural and functional diversity

A complete list of nematodes identified and total biomass values at each sampling site and at different sediment depth layers are reported in Table S2. Total biomass values ranged from 47.1 ± 8.7 to 149.3 ± 72.6 µg C at sites G and H, respectively. PERMANOVA analysis indicated that nematode biomass changed significantly between sites and along the vertical profile (Table 1). In details, the pair-wise test reported significant higher biomass values at site H compared to all the other sites and significant lower biomass values characterizing site G compared to the inactive sites. Moreover, nematode biomass significantly increased going deeper into the sediment and showing higher values at 3–5 cm and 5–10 cm layers compared to the top 3 cm at all sites except for CS (Table 1).

Nematode diversity indices (SR; ES$_{(51)}$; J), and functional diversity indices (ITD; MI) are reported in Table 2. Since we did not detect any significant differences in nematode composition with sediment depth layer (see below), we assessed for differences in diversity and functional indices only between sites (Table 1). Site H showed significant lower values

**Table 2 Indices of diversity at all sampling sites.**

|        | G1   | G2   | G3   | H1   | H2   | H3   | CN1  | CN2  | CN3  | CS1  | CS2  | CS3  |
|--------|------|------|------|------|------|------|------|------|------|------|------|------|
| SR     | 51   | 63   | 61   | 12   | 11   | 7    | 69   | 72   | 45   | 52   | 74   | 55   |
| ES(51) | 22   | 24   | 22   | 4    | 6    | 3    | 27   | 25   | 23   | 20   | 24   | 19   |
| J      | 0.80 | 0.80 | 0.77 | 0.29 | 0.28 | 0.34 | 0.85 | 0.81 | 0.86 | 0.73 | 0.71 | 0.67 |
| ITD    | 0.63 | 0.59 | 0.45 | 0.63 | 0.76 | 0.63 | 0.49 | 0.58 | 0.52 | 0.40 | 0.33 | 0.34 |
| MI     | 2.9  | 2.8  | 2.7  | 3.5  | 3.8  | 3.6  | 2.8  | 2.9  | 2.7  | 3.2  | 3.3  | 3.4  |

**Note:**
Nematode species richness (SR); the expected species number (ES[51]); Pielou index (J); the index of trophic diversity (ITD) and the maturity index (MI). All replicates are reported.

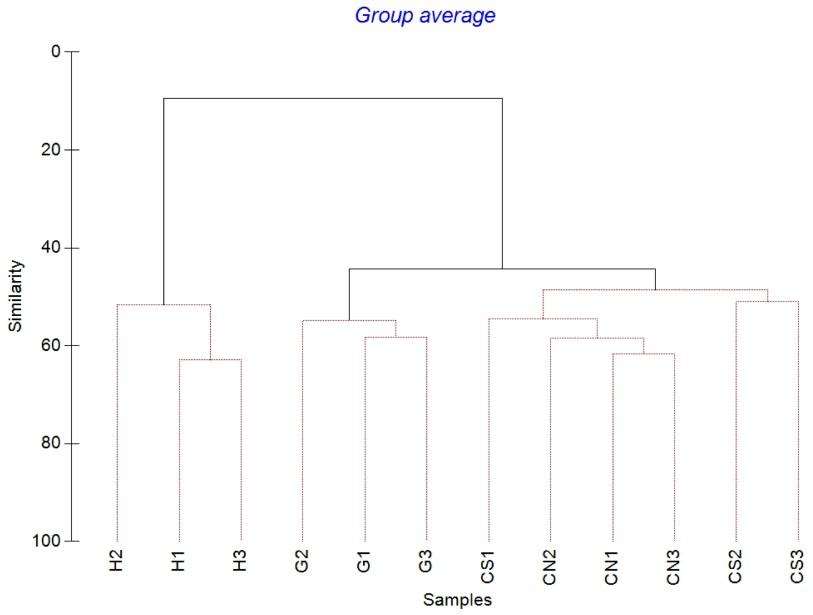

**Figure 7 Cluster analysis with SIMPROF test performed on nematode composition.** Red lines indicate statistically significant groupings according to the SIMPROF routine. All replicates are shown and indicated as: H1, H2 and H3; G1, G2 and G3; CN1, CN2 and CN3; CS1, CS2 and CS3.

in nematode diversity (SR; $ES_{(51)}$) and equitability (J′) compared to all the other sites; while the inactive site CS was characterized by significant lower values for J′ index compared to CN and G. Overall, a total of 33 families and 2 subfamilies; 156 genera and 165 species (i.e., morphotypes) were identified.

The PERMANOVA results showed significant effect of the factor "site" on nematode community composition (Table 1). The SIMPER analysis revealed a dissimilarity of 87–94% between site H and all other sites, of 55–57% between site G and inactive sites and of 50% between sites CN and CS (see also CLUSTER analysis, Fig. 7).

Among all identified families, the most diversified in term of number of genera were Desmodoridae (25 genera), Chromadoridae (21 genera), Cyatholaimidae (14 genera), Comesomatidae (11 genera) and Xyalidae (10 genera). Four families, represented only by one genus, were encountered just in one of the investigated sites: Ceramonematidae (site G), Siphonolaimidae (site H), Rhabdolaimidae and Rhadinematidae (site CS) (Table S2).

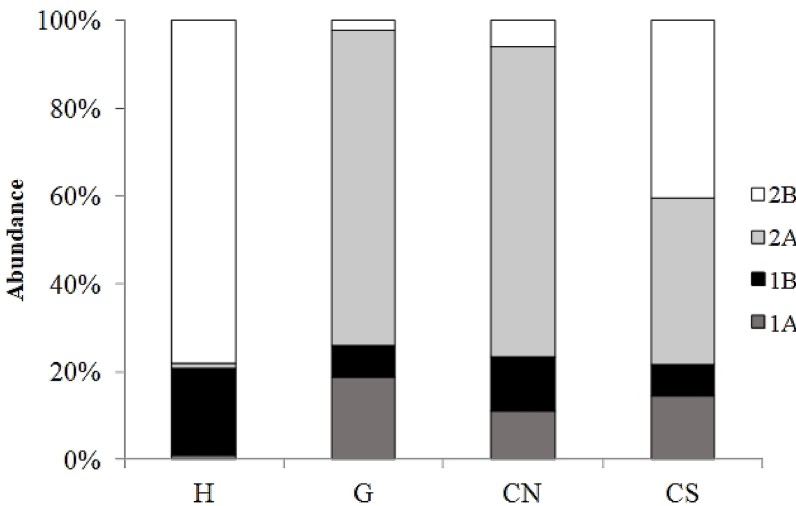

**Figure 8 Nematode trophic structure characterizing all investigated sites.** Reported are 1A (selective deposit feeders), 1B (non-selective deposit feeders), 2A (epigrowth feeders) and 2B (predators/omnivores) at the four sampling sites.

A total of 58 exclusive species were identified, all of them represented by few individuals (between 0.1% and 1.9% of the total abundances). The diffusive site H was characterized by the dominance of two nematode species *Oncholaimus* sp1 and *Daptonema* sp1 (less than 1% in the other sites) and by the presence of *Elzalia* sp1, *Linhystera* sp2 and *Parastomonema* sp1 as exclusive species. The geyser site G was characterized by the presence of five abundant species *Chromadorita* sp1, *Desmodora* sp1, *Leptolaimus* sp1, *Microlaimus* sp1 and *Paradesmodora* sp1 and by 22 exclusive species (Table S2).
The inactive sites were characterized by the presence of 16 (at CN) and 17 (at CS) unique species and by high abundances of *Desmodora* sp1, *Paradesmodora* sp1, *Perspiria* sp1 and *Spirinia* sp1 at CN site and by a slight dominance of *Chromaspirina* sp1 (33.4%) and a high abundance of *Spirinia* sp1 at CS (Table S2).

The trophic structure of nematode assemblages was dominated by predators/omnivores (2B; 78%) at site H and by epistrate feeders (2A) at sites G (72%) and site CN (71%); the inactive site CS was characterized by the co-dominance of the groups 2A (38%) and 2B (40%) (Fig. 8). Significant differences in the index of trophic diversity values between CS and all other sampling sites (Table 1) was reported. Values of the MI were significantly higher at site H and CS compared to the other sites (Tables 1 and 2).

## Relationship between meiofauna and environmental variables

The DistLM analysis was performed to assess the influence of environmental variables on faunal assemblages (i.e., meiofauna abundance, taxonomic composition, number of major taxa, nematode abundance and biomass, nematode composition and diversity and nematode trophic diversity). A combination of sediment (e.g., pH, $T\,°C$ and TOC content) and interstitial water (e.g., $NO_3^{2-}$, $Na^+$, $K^+$ and $S^{2-}$) variables considered in our investigation might explain the meiofauna and nematofauna variability in different percentage and depending on the meiofauna and nematode descriptors considered (Table 3).

Table 3 Results of DistLM procedure (Sequential test) for fitting sediment and interstitial water variables to the meiofauna and nematofauna variables.

| Meiofauna abundance | | | | | Meiofauna no. taxa | | | | | Meiofauna taxonomic composition | | | | | Nematode abundance | | | | |
|---|---|---|---|---|---|---|---|---|---|---|---|---|---|---|---|---|---|---|---|
| Variable | $R^2$ | $P$ | Var % | Cum % | Variable | $R^2$ | $P$ | Var % | Cum % | Variable | $R^2$ | $P$ | Var % | Cum % | Variable | $R^2$ | $P$ | Var % | Cum % |
| $Cl^-$ | 0.229 | ns | 0.23 | 0.23 | $NO_3^{2-}$ | 0.849 | 0.001 | 0.849 | 0.849 | $NO_3^{2-}$ | 0.475 | 0.001 | 0.475 | 0.475 | $NO_3^{2-}$ | 0.459 | 0.01 | 0.459 | 0.459 |
| $Zn$ | 0.280 | ns | 0.05 | 0.28 | $Mg^{2+}$ | 0.941 | 0.01 | 0.093 | 0.941 | $T$ (°C) | 0.582 | 0.03 | 0.107 | 0.582 | $K^+$ | 0.566 | ns | 0.107 | 0.566 |
| $T$ (°C) | 0.654 | 0.02 | 0.37 | 0.65 | pH | 0.951 | ns | 0.010 | 0.951 | pH | 0.664 | 0.03 | 0.082 | 0.664 | $T$ (°C) | 0.602 | ns | 0.036 | 0.602 |
| pH | 0.672 | ns | 0.02 | 0.67 | $Zn$ | 0.961 | ns | 0.009 | 0.961 | $Cu$ | 0.733 | 0.03 | 0.069 | 0.733 | $Na$ | 0.716 | ns | 0.114 | 0.716 |
| $Ca^{2+}$ | 0.678 | ns | 0.01 | 0.68 | $T$ (°C) | 0.964 | ns | 0.003 | 0.964 | $K^+$ | 0.785 | ns | 0.052 | 0.785 | $Zn$ | 0.732 | ns | 0.016 | 0.732 |
| $Pb$ | 0.820 | ns | 0.14 | 0.82 | $S^{2-}$ | 0.971 | ns | 0.007 | 0.971 | $Na$ | 0.817 | ns | 0.031 | 0.817 | $Ca^{2+}$ | 0.801 | ns | 0.070 | 0.801 |
| Mud | 0.928 | ns | 0.11 | 0.93 | TOC | 0.979 | ns | 0.008 | 0.979 | $Zn$ | 0.853 | ns | 0.036 | 0.853 | TOC | 0.901 | ns | 0.100 | 0.901 |
| $Cu$ | 0.946 | ns | 0.02 | 0.95 | $Ca^{2+}$ | 0.985 | ns | 0.007 | 0.985 | TOC | 0.921 | ns | 0.068 | 0.921 | $Cl^-$ | 0.993 | 0.004 | 0.092 | 0.993 |
| $Na$ | 0.994 | ns | 0.05 | 0.99 | $K^+$ | 0.992 | ns | 0.007 | 0.992 | $SO_4^{2-}$ | 0.964 | ns | 0.044 | 0.964 | $Mg^{2+}$ | 0.995 | ns | 0.002 | 0.995 |
| $S^{2-}$ | 0.999 | ns | 0.01 | 1.00 | $Pb$ | 0.999 | ns | 0.007 | 0.999 | $Ca^{2+}$ | 0.985 | ns | 0.021 | 0.985 | $Cd$ | 0.998 | ns | 0.003 | 0.998 |

| Nematode biomass | | | | | Nematode composition | | | | | Nematode diversity indices | | | | | Nematode trophic diversity | | | | |
|---|---|---|---|---|---|---|---|---|---|---|---|---|---|---|---|---|---|---|---|
| Variable | $R^2$ | $P$ | Var % | Cum % | Variable | $R^2$ | $P$ | Var % | Cum % | Variable | $R^2$ | $P$ | Var % | Cum % | Variable | $R^2$ | $P$ | Var % | Cum % |
| $K^+$ | 0.630 | 0.001 | 0.630 | 0.630 | $NO_3^{2-}$ | 0.521 | 0.001 | 0.521 | 0.521 | $NO_3^{2-}$ | 0.888 | 0.01 | 0.888 | 0.888 | $Cl^-$ | 0.627 | 0.01 | 0.627 | 0.627 |
| $Cu$ | 0.699 | ns | 0.069 | 0.699 | $Na$ | 0.633 | 0.001 | 0.112 | 0.633 | $T$ (°C) | 0.939 | 0.02 | 0.050 | 0.939 | TOC | 0.856 | 0.01 | 0.229 | 0.856 |
| $NO_3^{2-}$ | 0.731 | ns | 0.032 | 0.731 | $Ca^{2+}$ | 0.709 | 0.003 | 0.076 | 0.709 | $S^{2-}$ | 0.976 | 0.003 | 0.038 | 0.976 | $T$ (°C) | 0.882 | ns | 0.026 | 0.882 |
| Mud | 0.742 | ns | 0.012 | 0.742 | TOC | 0.757 | ns | 0.048 | 0.757 | $Mg^{2+}$ | 0.980 | ns | 0.004 | 0.980 | $Cu$ | 0.920 | ns | 0.039 | 0.920 |
| $Zn$ | 0.858 | ns | 0.116 | 0.858 | $T$ (°C) | 0.798 | ns | 0.042 | 0.798 | $Zn$ | 0.983 | ns | 0.002 | 0.983 | $NO_3^{2-}$ | 0.964 | 0.03 | 0.043 | 0.964 |
| TOC | 0.938 | ns | 0.080 | 0.938 | $Pb$ | 0.837 | ns | 0.039 | 0.837 | $Na$ | 0.985 | ns | 0.003 | 0.985 | $Cd$ | 0.973 | ns | 0.010 | 0.973 |
| pH | 0.982 | 0.01 | 0.044 | 0.982 | $Cu$ | 0.872 | ns | 0.035 | 0.872 | $K^+$ | 0.994 | 0.03 | 0.008 | 0.994 | $Na$ | 0.985 | ns | 0.012 | 0.985 |
| $Ca^{2+}$ | 0.987 | ns | 0.005 | 0.987 | $Mg^{2+}$ | 0.906 | ns | 0.035 | 0.906 | $Ca^{2+}$ | 0.997 | ns | 0.003 | 0.997 | $Mg^{2+}$ | 0.989 | ns | 0.004 | 0.989 |
| $Cl^-$ | 0.991 | ns | 0.004 | 0.991 | $Cl^-$ | 0.944 | ns | 0.038 | 0.944 | $Cu$ | 0.998 | ns | 0.001 | 0.998 | $K^+$ | 0.994 | ns | 0.005 | 0.994 |
| $Mg^{2+}$ | 0.998 | ns | 0.007 | 0.998 | pH | 0.977 | ns | 0.033 | 0.977 | $Pb$ | 0.999 | ns | 0.001 | 0.999 | $SO_4^{2-}$ | 0.997 | ns | 0.003 | 0.997 |

Note:
% Var, percentage of explained variance; % Cum, cumulative percentage explained by the added variable; ns, not significant.

The dbRDA graph (Fig. 9) on meiofauna taxonomic composition showed a separation between sites due to changes in $T$ °C (higher at H site), pH (more acid at H site) and $NO_3^{2-}$ and Cu concentrations. All these variables could explain the 73% of variability in meiofauna composition (Table 3). The dbRDA graph (Fig. 10) on nematode composition showed that changes in nematode communities were significantly correlated (71%) to interstitial water features (Table 3) with a separation of the sampling sites into three main groups: H, G and the inactive sites CN–CS.

# DISCUSSION

## Small-scale spatial environmental heterogeneity induced by vent emissions

Environmental conditions at hydrothermal vent systems markedly differ from background and these differences include increase of temperature, decrease of pH and enrichment in

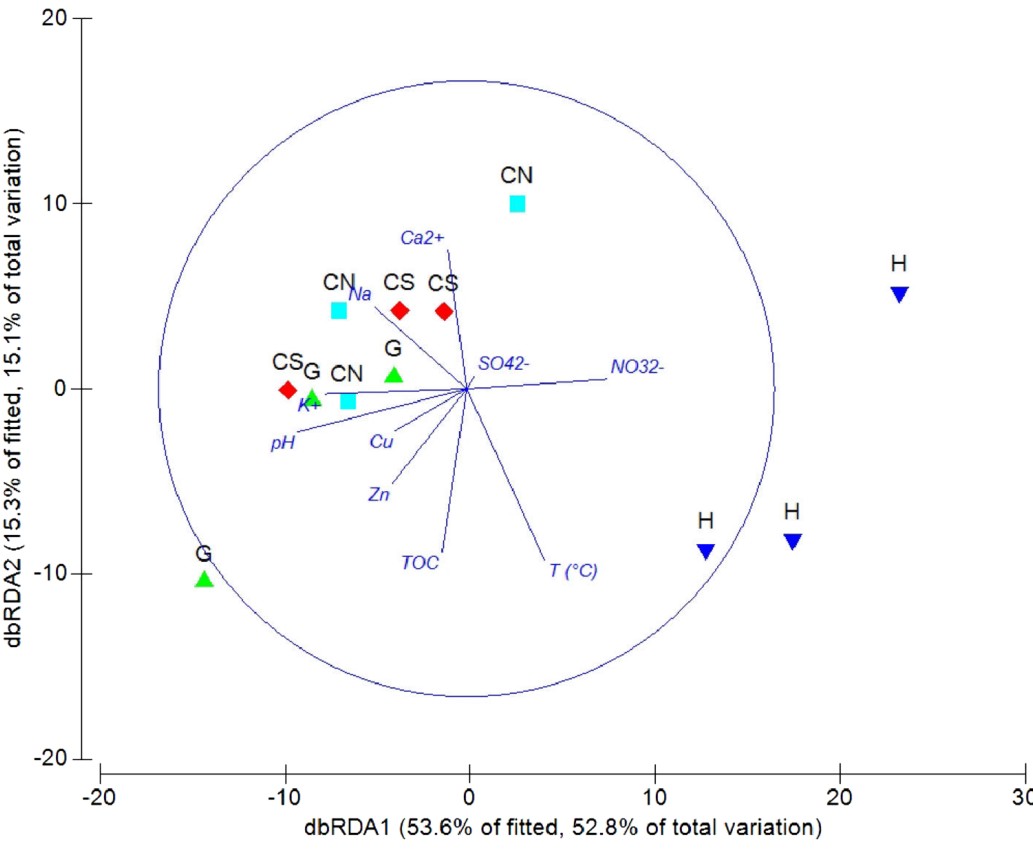

**Figure 9 dbRDA canonical model on meiofauna taxonomic composition.** The graph shows the effect of environmental variables on meiofauna taxonomic composition based on Spearman rank correlations.

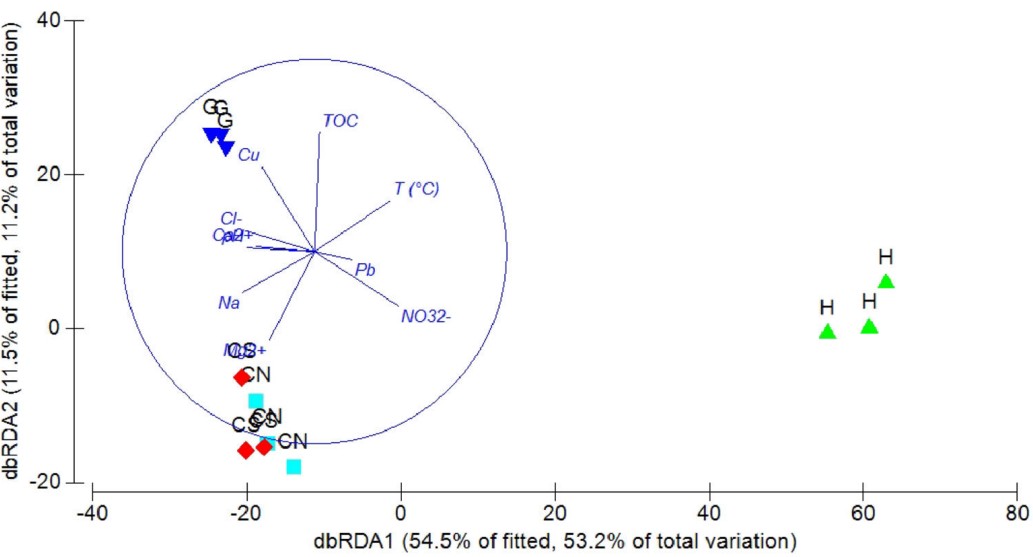

**Figure 10 dbRDA canonical model on nematode community composition.** The graph shows the effect of environmental variables on nematode community composition based on Spearman rank correlations.

reduced compounds, gases and heavy metals (*Tarasov et al., 2005*). Shallow and deep vents are characterized by high heterogeneity and can be defined as "patchy habitats" (*Gollner et al., 2010*). Due to hydrothermal activity, patchiness has often been detected in physical and chemical variables at small spatial scales (*Dando et al., 1995*; *Melwani & Kim, 2008*, *Gollner et al., 2010*). Similarly, at the shallow vent area of SdF we detected a rapid change in the environmental conditions with a consequent marked environmental heterogeneity on a relatively small spatial scale (i.e., 100 m).

SdF is characterized by three different habitats: (1) the diffusive emission site (H) in which we reported the highest temperature, acidic conditions as a result of $CO_2$ gas bubbling (*Di Napoli et al., 2016*) and the presence of a white microbial mat; (2) the solitary geyser site (G) which differed from the other sites for the presence of sulfur ions in the interstitial waters and yellow sulfurous deposits, medium–high temperature conditions but ambient pH values and (3) the inactive sites (CN and CS) characterized by no extreme conditions (T and pH values within the natural range), the highest percentage of gravel fraction and the lowest TOC content compared to the active sites.

## Meiofauna distribution and taxonomic composition variability at vent conditions

In the deep-sea, vent emissions negatively impact fauna abundance (*Kamenev et al., 1993*; *Tarasov et al., 1999*, *2005*), while in shallow water the amount of food material derived from both chemosynthesis and photosynthesis processes (*Sorokin, Sorokin & Zakouskina, 2003*) promote the meiofauna abundance that can reach higher values compared to the background sediments (*Kamenev et al., 1993*).

At SdF vent area active and inactive sites did not differ significantly in total meiofauna abundance, contrarily to the drastic drop registered for the macrobenthic density at active sites compared to the inactive sites at SdF (*Donnarumma et al., 2019*). Overall, total meiofauna abundance reported in this study were higher than those at other shallow vent areas (*Kamenev et al., 1993*; *Thiermann et al., 1997*; *Tarasov et al., 1999*; *Colangelo et al., 2001*; *Zeppilli & Danovaro, 2009*). The lack of significant differences between sites is possibly due to a pronounced inter-replicates variability that reflects the irregular and patchy distribution of meiofauna. We found the same small spatial-scale patchiness in the vertical distribution of meiofauna, with no significant differences in total abundances moving deeper into the sediment layers but only a trend in increasing abundance values from the surface to the deeper layers. Meiofauna is known to be characterized by a patchy distribution and this is particularly evident at small spatial scale (i.e., microhabitats; *Gallucci et al., 2008*). At vents, the very small-scale distribution is related to the spatial heterogeneity of biogeochemical vent processes that normally occurs in these extreme environments (*Tarasov et al., 2005*; *Di Bella et al., 2016*). In this study, the combination of vent small heterogeneity and meiofauna patchiness is particular evident.

Meiofauna distribution can vary more along the sediment vertical profile than along a horizontal axe (*Fonseca et al., 2010*). Only few studies documented the vertical

distribution of meiobenthic organisms inhabiting shallow vent areas showing a decreasing trend with increasing sediment depth (*Ansari, Rivonkar & Sangodkar, 2001*; *Di Bella et al., 2016*). Sediment physical disturbance is known to impact meiofaunal vertical distribution (*Braeckman et al., 2011*). In some cases, nematodes can migrate from the surface disturbed layers into the less impacted deeper layers (*Leduc & Pilditch, 2013*). At SdF the highest abundances were recorded at the deeper layers. The bubble streams coupled with near-bottom currents and sediment rework by larger fauna might possibly have a role in meiofauna sediment penetration by inducing a migratory response to the deeper layers (*Colangelo et al., 2001*; *Steyaert et al., 2003*).

Meiofauna diversity was higher at SdF than in the other shallow vent meiofauna studies (*Fricke et al., 1989*; *Kamenev et al., 1993*; *Dando et al., 1995*; *Tarasov et al., 1999*; *Colangelo et al., 2001*; *Zeppilli & Danovaro, 2009*). Nematodes represented a key taxon of meiobenthic communities at shallow vents (*Kamenev et al., 1993*; *Thiermann et al., 1997*; *Tarasov et al., 2005*). Similarly to results of *Colangelo et al. (2001)* and *Zeppilli & Danovaro (2009)*, we reported a dominance of copepods at one of our four sites, site H, which showed a different assemblage structure in respect to the others. Previous authors (*Coull, 1985*; *Colangelo et al., 2001*) explained this dominance invoking the alteration of sediment grain size and the preference for coarser sediments by this taxon, coupled with the effect of intermediate disturbance created by a moderate gas bubbling that seemed to promote copepod diversity (*Colangelo et al., 2001*). At the diffusive emission site (H), we did not detect any clear alteration of sediment grain size but we can hypothesize a gas bubbling effect due to the presence of $CO_2$ emissions. Moreover, the presence of white microbial mat at site H could provide food for vent obligate copepods, which have developed specific adaptation to feeding on bacterial mats (*Heptner & Ivanenko, 2002*). High abundance of copepods characterized by a high SR was reported also from deep-sea vents (*Gollner et al., 2010*), confirming their ability to be well adapted to the extreme vent conditions. Two other taxa, usually neglected in the meiobenthic studies because part of the unicellular organisms, constituted two abundant components of site H vent assemblage and they probably took advantage feeding on microbial mat: foraminifers and ciliates. *Bernhard et al. (2000)* found that the microbial mats in the anoxic, sulphidic waters of the Santa Barbara Basin supported abundant communities of foraminifera, flagellates and ciliates. Benthic foraminifers represent an important environmental sensitive group and their composition is strictly related to venting activity (*Di Bella et al., 2016*). Protists (e.g., ciliates) play several roles in marine ecosystems and form a trophic link between prokaryotes and higher trophic levels; they impact carbon and other nutrient cycles directly and indirectly through grazing on organic matter and prokaryotic prey (*Anderson, Winter & Jürgens, 2012*).

The meiofaunal taxonomic composition usually varies between surface and subsurface sediment layers: copepods and *nauplii* occupy the well oxygenated surface sediment layer (*Grego et al., 2014*), while nematodes become the dominant taxon at subsurface depths (*Ingels et al., 2009*; *Rosli et al., 2016*). Meiobenthic organisms at SdF vent followed the same trends for nematodes and copepods at all sites.

## Nematode community reflects Secca delle Fumose environmental heterogeneity

At SdF shallow vent, nematode community composition and functional diversity varied from site to site reflecting the environmental heterogeneity of the sampling area. The same marked change in fauna composition was reported for the macrobenthic communities inhabiting SdF, with an absence of more sensitive species at the active sites (*Donnarumma et al., 2019*). The high variability in the nematode assemblages (i.e., percentage of dissimilarity between sites) and the presence of exclusive species characterizing each sampling site accounted for the differences in the biodiversity composition. This change in nematofauna was mainly due to a decrease of nematode genera/species at site H and to a change in the abundances and/or replacement of species among the other sites (G vs. CN vs. CS).

At the diffusive emission site H, where we reported the most extreme conditions, nematode assemblage showed some typical traits of nematode assemblages inhabiting shallow (*Thiermann, Windoffer & Giere, 1994*; *Dando et al., 1995*; *Tarasov et al., 1999*; *Zeppilli & Danovaro, 2009*) and deep-sea active vent sites (*Tchesunov, 2015*; *Zeppilli et al., 2018*): biomass values from twofold to threefold higher compared to that of the active site G and inactive sites due to the dominance of big size *Oncholaimus* genus (i.e., very low equitability value), the lowest nematode diversity (SR) and functional diversity (trophic diversity). In this case, and as reported in *Donnarumma et al. (2019)* for the macrofauna, the harsh hydrothermal conditions affect the nematode assemblages.

Nematodes belonging to the family Oncholaimidae (genus *Oncholaimus*) have been reported several times inhabiting sediments near to source of the emissions at shallow vents (*Dando et al., 1995*; *Thiermann et al., 1997*; *Zeppilli & Danovaro, 2009*). This genus can tolerate high sulfide concentrations by producing sulfur-containing droplets to reduce the toxic effect of hydrogen sulfide (*Thiermann, Windoffer & Giere, 1994*; *Thiermann et al., 1997*). *Oncholaimus* has been originally identified as predator/omnivore/scavenger (*Jensen, 1987*), but recently it has been shown that this nematode can feed also on free-living chemoautotroph microorganisms (*Zeppilli et al., 2019*). This wide diet makes *Oncholaimus* able to use the different food sources available at shallow vent: other organisms (as alive or dead animals) and bacterial mat (*Thiermann et al., 1997*). Oncholaimidae are also known to have symbiotic associations (*Bellec et al., 2018*, *2019*). The deep-sea nematode *Oncholaimus dyvae* can harbor sulfur-oxidizing bacteria in the cuticle and in its intestine (*Bellec et al., 2018*). *Metoncholaimus albidus* inhabiting shallow water anoxic sediments hosts ectosymbiotic bacteria involved in sulfur metabolism suggesting a potential for chemosynthesis in the nematode microbial community (*Bellec et al., 2019*). At SdF, *Oncholaimus* showed endosymbiotic sulfur-oxidizing and -reducing bacteria, purple sulfur bacteria and *Zetaproteobacteria* in the intestine (L. Bellec et al., 2019, Unpublished data).

Moreover, the high abundance and biomass values characterizing *Oncholaimus* suggested that this nematode could have a significant impact on the turnover of organic matter. *Daptonema* sp1 was the second most abundant species at site H; this genus was

frequently found at vent areas (*Vanreusel, Van Den Bossche & Thiermann, 1997*; *Zeppilli & Danovaro, 2009*). The low trophic diversity index value was the consequence of the dominance of the genera *Oncholaimus* and *Daptonema*, but deposit feeders and predators/scavengers are typically reported from shallow vent systems where organisms can feed on available organic resources.

The drastic decrease in macrobenthic biodiversity observed at geyser site due to the effect of high sediment temperature and sulfide ions (*Donnarumma et al., 2019*), was not reported for the nematofauna that showed very low diversity values only at H site.

The nematode diversity at geyser (site G) was comparable to that at the inactive sites, even if the geyser hosted its own nematode community characterized by the highest number of exclusive species (22). While the macrobenthic assemblages around site G was defined as "simplified" community that represented a subset of the background biota (*Donnarumma et al., 2019*), the presence of sulfur ion $S^{2-}$ did not constitute a source of disturbance for nematodes structural and functional diversity. According to the ecological theory of "intermediate disturbance" (*Huston, 1979*), across an environmental stress gradient higher diversity is expected at intermediate stress levels (i.e., site G), whilst at higher levels of stress only colonizing species survive (i.e., site H). Similarly, *Colangelo et al. (2001)* reported higher copepod diversity values in areas with moderate gas seepage and sulfur deposits. At site G we did not observe a clear dominance of one or few genera (i.e., high value of equitability index), and the most represented genera such as *Leptolaimus, Desmodora, Paradesmodora* and *Chromadorita* were already reported from other vent areas (*Dando et al., 1995*; *Vanreusel, Van Den Bossche & Thiermann, 1997*; *Zeppilli & Danovaro, 2009*).

At inactive sites (CN and CS) the family of Desmodoridae (genera *Spirinia, Perspiria* and *Chromaspirina*) was the most represented taxon. Those genera, and in particular different species of *Spirinia* and *Chromaspirina*, have been reported from different environments spanning from shallow (*Platt, 1977*; *Nicholas et al., 1991*; *Ólafsson, 1995*; *Vafeiadou et al., 2014*) to deep-sea (*Da Silva et al., 2009*; *Leduc & Verschelde, 2015*) systems and sometimes associated to unstable condition and coarser sediment, as the sediment at our inactive sites from where we detected the highest percentage of gravel. Another group found at inactive sites was that of Stilbonematinae, reported already from shallow (*Kamenev et al., 1993*; *Thiermann, Windoffer & Giere, 1994*; *Dando et al., 1995*) and deep-sea vent areas (*Vanreusel, Van Den Bossche & Thiermann, 1997*; *Gollner et al., 2010*) and inhabiting zones of volcanic activity or at the periphery of vents. Stilbonematine nematodes are common in suitable tropical shallow-water carbonate sands (*Ott, Bright & Bulgheresi, 2004*), but they are also adapted to life at seeps, organically enriched bottoms (*Giere, 2009*) and sulphidic sediments where they feed on ectosymbiotic sulfur-oxidizing bacteria (*Ott et al., 1991*). Stilbonematine nematodes have chemosynthesising ectobacteria covering their bodies, which constitute an adaptation to life in silt enriched with $H_2S$ (*Powell, Crenshaw & Rieger, 1980*). We found stilbonematine nematodes at SdF sediments with their body completely covered by ectobacteria. Their presence at inactive sites, even if in low abundances, suggests the
presence of localized and very likely patchy reduced conditions at those sites (*Tchesunov, Ingels & Popova, 2012*) caused by accumulation and burial of dead organic matter.

The trophic structure of assemblages inhabiting the geyser and the inactive site CN was similar to that reported from other shallow vent systems (*Kharlamenko et al., 1995*; *Zeppilli & Danovaro, 2009*), were epistrate feeders dominated because favored by the high primary biomass. The bulk of biomass does not rely solely on symbiotrophs, but on the available organic resources (i.e., deposit feeders, epistrate feeders, predators/omnivores) and this was confirmed also by the significant correlation between TOC content and nematode trophic diversity at SdF. This is particularly true for the inactive site CS where all trophic groups were well represented (i.e., the lowest value of ITD) and therefore underling the ability of nematodes to use all the available food sources and /or to partition multiple food sources (*Limén, Levesque & Juniper, 2007*).

Vent nematodes belong to families and genera already known from non-vent habitats (*Vanreusel, Van Den Bossche & Thiermann, 1997*; *Flint et al., 2006*; *Zeppilli et al., 2018*), suggesting no endemicity at genera or families level for this taxon. However, at a species level many of the nematode present in samples from deep-sea vents appeared to be new, suggesting the presence of unique or endemic species adapted to the vent conditions (*Copley et al., 2007*; *Gollner et al., 2010*). In contrast to deep-sea hydrothermal vents, nematode inhabiting shallow vent areas include a subset of species that live in background sediments but can survive in extreme conditions (*Tarasov et al., 2005*; *Zeppilli & Danovaro, 2009*; *Zeppilli et al., 2018*). However, this is was not the case at SdF vent area where each one of the investigated site (i.e., active H–G and inactive CN–CS sites) was characterized by a distinct nematode community reflecting the high spatial heterogeneity. The 26% (at site CN) and 28% (at site CS) of species were unique at the inactive sites, but the highest percentages of exclusive species were found characterizing the active sites H (30%) and G (38%). In our study, we noticed a lack of dominance by the nematode genera usually found to be abundant at shallow (e.g., *Sabatiera, Linhomoeus, Siphonolaimus, Pomponema, Dichromadora, Paracanthoncus* and *Steineridora*; *Dando, Hughes & Thiermann, 1995*; *Zeppilli & Danovaro, 2009*) and deep-sea (*Molgolaimus, Monhystera* and *Thalassomonhystera*; *Vanreusel, Van Den Bossche & Thiermann, 1997*; *Zekely et al., 2006*) vents, except for the genus *Oncholaimus*. On the other hand, most of the exclusive genera/species inhabiting the investigated sites are not frequently reported from other vent areas, sustaining the presence of a typical nematofauna assemblage at SdF. Moreover, we noticed a preponderance of monospecific nematode genera in all samples with the 95% of genera that exhibited a species (i.e., morphotype) to genus ratio of 1 as reported from deep-sea vent systems (*Vanreusel, Van Den Bossche & Thiermann, 1997*; *Gollner et al., 2010*).

## Effects of SdF vent conditions on meiobenthic comunities

In the recent study of *Donnarumma et al. (2019)*, it has been reported how macrobenthic community structure was significantly correlated with some of the environmental variables (e.g., pH, TOC, temperature and interstitial water ions), the same variables that mostly

determined the differences between the investigated sites. Similarly, the meiofauna and nematofauna richness and community composition were correlated with the same environmental parameters (Table 3). The effects of sediment temperature, TOC, pH and ions (e.g., $NO_3^{2-}$, $Ca^{2+}$ and $Cl^-$) were particularly clear in influencing patterns of meiofauna and nematofauna communities (Figs. 7 and 8).

Separation between H, G and CN–CS sites was even more evident when we considered the nematode community composition. The same separation between sites occurred if we considered the macrobenthic community composition (*Donnarumma et al., 2019*), confirming the importance of key environmental factors for benthic communities such as: sediment temperature, pH value, TOC content and interstitial water characteristics. But if for the macrobenthic community those key factors led to a drastic decrease in density and diversity at both active sites, for the meiofauna and nematofauna assemblage this effect was reported only at H site where conditions were the most extreme and were few adapted species can survive.

## CONCLUSIONS

This study confirmed some of the trends often observed in vent-associated benthic communities, that is, a pronounced small-scale spatial variability of meiofauna and particularly nematode composition that reflect the natural patchy distribution of this benthic components and the high environmental heterogeneity of the study area, typically from extreme environments. We noticed a migratory response to deeper layers by the meiofauna due to sediment disturbance (e.g., bubble streams, near-bottom currents, sediment reworking by larger fauna), this is also a common phenomenon reported from such a kind of environment. High nematode biomass values, low diversity and the dominance by the single highly tolerant genus *Oncholaimus* at station with the harshest conditions (H) was not surprising for vent systems. Other findings, however, appear to contradict some general accepted tenets of shallow water hydrothermal vents ecology. We reported higher values of meiofaunal abundance and diversity characterizing SdF shallow vent compared to other shallow vent areas. Nematodes inhabiting sediments of SdF were clearly different and each one of the investigated site was characterized by a distinct nematode community, that means that nematodes at SdF shallow vent did not constitute a subset of species that live in background sediments. In our study, we noticed a lack of dominance by the nematode genera usually found to be abundant at shallow and deep-sea vents, except for *Oncholaimus* genus, sustaining the presence of a typical nematofauna community at SdF.

## ACKNOWLEDGEMENTS

We thank the Soprintendenza of the Underwater Archeological Park of Baia (prot. 5667, 24/10/2016) for the authorization to sampling. We are in debt to the diving center Centro SUB Pozzuoli (Guglielmo Fragale) for support during diving activities. We thank Dr. Renato Bruno, Dr. Aurélie Tasiemski and Dr. Céline Boidin-Wichlacz for helping in sampling activities.

### Funding

This study was supported by the project "Prokaryote-nematode Interaction in marine extreme envirONments: a uniquE source for ExploRation of innovative biomedical applications" (PIONEER) funded by the Total Foundation and IFREMER (2016–2019) and by the project Boost Europe – ERC "Extreme marine nematoDes: moDel orgAnisms for a Journey toward the origin and Evolution of metazoan life in our changing planet" (EDDAJE) funded by the Brittany Region and IFREMER (2019). The funders had no role in study design, data collection and analysis, decision to publish, or preparation of the manuscript.

### Grant Disclosures

The following grant information was disclosed by the authors:
Total Foundation and IFREMER (2016–2019).
Brittany Region and IFREMER (2019).

### Competing Interests

The authors declare that they have no competing interests.

### Author Contributions

- Elisa Baldrighi performed the experiments, analyzed the data, prepared figures and/or tables, authored or reviewed drafts of the paper, and approved the final draft.
- Daniela Zeppilli conceived and designed the experiments, performed the experiments, analyzed the data, authored or reviewed drafts of the paper, and approved the final draft.
- Luca Appolloni performed the experiments, analyzed the data, prepared figures and/or tables, authored or reviewed drafts of the paper, and approved the final draft.
- Luigia Donnarumma performed the experiments, analyzed the data, prepared figures and/or tables, authored or reviewed drafts of the paper, and approved the final draft.
- Elena Chianese analyzed the data, authored or reviewed drafts of the paper, and approved the final draft.
- Giovanni Fulvio Russo analyzed the data, authored or reviewed drafts of the paper, and approved the final draft.
- Roberto Sandulli conceived and designed the experiments, performed the experiments, analyzed the data, authored or reviewed drafts of the paper, and approved the final draft.

### Data Availability

    The raw measurements are available in the Supplemental Files.

### Supplemental Information

Supplemental information for this article can be found online at http://dx.doi.org/10.7717/peerj.9058#supplemental-information.

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
