# Peer review of "Meiofaunal communities and nematode diversity characterizing the Secca delle Fumose shallow vent area (Gulf of Naples, Italy)"

_PeerJ, doi:10.7717/peerj.9058_

## Round 0.1 · original submission · Minor Revisions

Reviewers have pointed out several issues that require further attention, otherwise, this looks to be an important contribution to the field of hydrothermal vent biology.

·

Basic reporting

The paper is about the characterization of meiofauna community in shallow hydrothermal vent in the gulf of Naples. The argument appears original an the work deserves to be published, but several inconsistences and corrections must be made before. Several concerns emerged by the images and table formatting. The numeration of figures as reported in the MS does not reflect the actual numeration of the single images reported at the end of the manuscript. The captions must be reformulated to be more informative in cases the images and tables are extrapolated by the text (they should be autoinformative). Problems with some references formatting emerged as well. Moreover, it seems that conclusion is lacking.

Experimental design

The experimental design is good for the questions the authors wanted to give an answer. The sampling appears rigorous as well. The analyses performed are quite standard and well applied.

Validity of the findings

Following the good experimental design, also the findings are valid and well interpreted.

Additional comments

Following I report several points that the author should modify.
Introduction
Line: 85: a dot at the end of the sentence seems to lack. “…for this taxon. However….”
Lines 112-122: Please move this part to the study area description or to the discussion. In the introduction report only the background problem of study shallow vents.
Material and Methods
Lines 139-144: move all this part to the sampling strategy.
Line 147: delete the reference as it is useless.
Lines 148-149: can you specify how did you collect the sediment? By scuba diving or what else?
Data analysis.
I would rearrange all this part according to the following suggestions:
Line 189: at the end of the sentence, please report “Faunal data were log(x+1) transformed and analysed using tests based on Bray-Curtis similarity matrices (multivariate analyses) and Euclidean similarity matrices (univariate analyses).”. Delete the lines 197-198 afterwards.
Lines 192-193: move this sentence after the Permanova description one. Moreover, change the sentence as “Pair-wise tests were carried out to verify the significance of the differences among sites and layers if any were observed in the main test.”.
Line 200: at the end of the sentence add “….(using 70% as cut-off) to determine the contributions of each species (cit).”. Delete lines 202-204 (comprising the sentence “This analysis of dissimilarities…”.
Line 201: delete between brackets “e.g. based…” leave only the reference.
Line 205: can you provide more info about the cluster technique you adopted? K-means? UPGMA?
Line 207: delete for.
Lines 209-217: something went wrong with the rows formatting.
Line 208: after brackets write “with a forward selection of the independent variables and 4999 permutations of residuals”. Delete lines 212-218.
Line 219: write “All the analyses were carried out by means of the software….”.
Results.
Lines 223-235: are these analyses made by you or by Donnarumma et al? If the last is true, please report these info in the study area paragraph and report in here only your own results.
Line 240: at least the first time please what the numbers are. For example, 1142±713.8 ind/cm2 (mean ± standard error, hereafter). Check the figure numbers reported in the MS. There is not a fig 2a. Check all the figure numeration all along the text and be consistent with actual figure numbers.
Line 244: delete (Permanova, not significant) you already said it was not significant, it is a useless repetition. Report just the table where one can found the test. It is easier for the reader. Moreover, in the tables report the actual p value to give the readers the possibility to interpret on their own the statistical significance of the test, above all in these times of “significance” meaning debate.
Line 247: delete these brackets, here and all along the MS.
Line 252: what the numbers are? Please report the units.
Lines 253-255: In this sentence saying that the site was the most important factor is not totally correct. In tests like this, with interactions, the interaction itself is the most important (although this terminology is inappropriate), and you, as a rule of thumb, must be look first at that term. Sometimes you can even ignore the significance of the single term significance in case the interaction is significant. So, discuss the interaction and the corresponding pairwise, rather than the single term.
Line 256: in brackets report the actual p value. Better, report the pairwise in the same table of the main tests or in a separate table. This applies for all the pairwise in the MS.
Line 262: check figure.
Line 265: change “was” with “were”.
Line 284: again report actual p value or table, hereafter.
Lines 295-296: The ANOSIM Is not reported in the MM. I suggest to delete this sentence, even because the results are redundant with PERMANOVA.
Line 298: I don’t understand which is the correspondent figure 5.
Lines 317-319: Here it is not clear if the main test is reported or the pairwise. If the last is true, where the main test is?
Line 328: Something went wrong with table 3 formatting please reduce the width of the columns to be sure the entire table is within the margins.
Discussion
Line 350: close brackets after 2016.
Line 544: A summary of the conclusion is lacking. Can you summarize all the findings and their ecological importance in a proper paragraph?
References
Check the formatting of the references of the following lines: 613, 649. The journal names should be in italic. Make a general check of the PeerJ formatting standards all along the MS. I suggest for references to use bibliographic software.
Figure and table captions.
Please, be sure to report the legend for all terms in the figures and tables. Imagine the figures as standalone products that should be auto informative if extrapolated from the MS.
Figure 1: I don’t like this image, but I understand you wanted to maintain it because it is part of complex work and was reported also in Donnarumma et al. But at least, give a more detailed description of the image. For example, does the image be captured with a probe like a scan sonar or what else?
Figure 9 caption: please delete “DistLM (distance-based linear model)” report only “dbRDA canonical model….”. Moreover, what do the segments correspond to? Correlations (Pearson or Spearman) with axes? Or what else? Same comments for the next figure 10.

Reviewer 2 ·

Basic reporting

Clear and unambiguous, however further check is needed for consitency and definitions (see comments below).
Literature references, sufficient field background/context provided.
Professional article structure, figures, tables. Raw data shared.
Self-contained with relevant results to hypotheses.

Experimental design

Original primary research.
Research question well defined, relevant and meaningful.
Original and novel.
Rigorous investigation performed to a high technical and ethical standard.
Methods described with sufficient detail and information to replicate.

Validity of the findings

All underlying data have been provided; they are robust, statistically sound and controlled.
Conclusions are well stated, linked to original research question and limited to supporting results.

Additional comments

General comments
The paper provides new insight on a very interesting ecological aspect which has been also poorly investigated so far. In addition, the experimental design and the data treatment are rigorous and ecologically sounding. So that, this article could be very interesting for the wide audience of PeerJ. However, I suggest to the Authors to carefully check the entire ms, since I am confident that it could be even ameliorated. First of all, I suggest to check the entire ms for consistency. Authors should maintain the same logic flow throughout the whole ms. More specifically, they should talk about characteristics of meiofaunal assemblages in shallow vs deep vents, characteristics of nematode assemblages in shallow vs deep vents, at genera and species level, and meiofauna vs macrofauna in vents. When they talk about “diversity” they should explore first the “level of biodiversity”, in terms of richness of taxa and diversity indexes, and then explore the putative differences in taxonomic composition. Following this scheme, they would avoid the missing of some important information (see comment on the Discussion section below). I would suggest also to revise and check the terms used throughout the ms, first for their definition in theoretical ecology (e.g., definition of “population”) and then for their consistency throughout the entire ms. This could help the reader in following the logical framework.
Below my specific comments.

Introduction
The Introduction is well written and quite clear, but maybe could be simplified in some parts. Some paragraphs seem very detailed and the specific information could be reported in the Discussion section. See as example, lines 83-88.
Line 103. A population is composed by individuals belonging all to the same species. I would change into “assemblage” and use the terms “assemblage” throughout the entire ms.
Lines 83-88 and 114-122. I would move this paragraphs to the Discussion section. The Introduction should just present the ecological problem and hypothesis and not provide all the available information.

Methods
It is unclear how sediment cores have been collected. By scuba diving?
Line 145. Those described are sampling methods and not strategy. The sampling strategy has been already described in the previous paragraph.
Line 165. “Until” indicates a temporal sequence.
Line 171. I would use “site” instead “station”, in order to be consistent throughout the entire ms.

Data analysis
Line 190. “Descriptors of meiofauna and nematode assemblage composition” is unclear. Which are they? And more specifically, which are the descriptors of assemblage composition?
Line 192: Pair wise tests are not for verifying the significance of differences, but to ascertain specifically between which samples the significant differences are.
Line 195. I suggest to calculate the diversity indexes cumulatively for the 3 replicates and describe the pattern, avoiding the statistical analysis. Indeed, using separatly the replicates could be misleading in the idenx caluclation, since in this way the index do not consider the species in common/exclusive among the 3 replicates. So that, the results of the statistical tests could be deviated. Otherwise, I suggest to maintain both approaches (expressing the indexes as avg+st.dev carrying out statistical tests and calculating them also cumulatively).
Line 196. Replace “communities” with “assemblages”.
Line 207. Replace “community composition” with “species composition”.
Line 211. Replace “diversity and community composition” with “richness of taxa and taxonomic composition”. The term “diversity” is too vague here, it is unclear what Authors are referring to. According to the ecological definition of “community” (different species living in the same area), meiofauna are not a community (because meiofauna are not identified at species level), rather an assemblage.
Line 212. Please, replace with “nematode trophic diversity and life strategies”.
Line 218. PCA has not been performed in this paper. Maybe dbRDA? Also ANOSIM is missing here? Please, check.

Results
Line 229. Why temperature is not expressed as avg±st.dev. here?
Line 232. Maybe “surrounding environment” is better?
Line 237. “Meiofauna abundance, community composition and distribution”. Please, replace with “taxonomic composition”.
Line 238. How a composition can be reported in a table? I would say "abundance of each taxon".
Line 241. As far as I understood these differences are not statistically significant.
Line 249. “5_10”? I would use 5-10 cm, as in other sections of the doc.
Line 252. See comment above on the diversity indexes. Same for the richness of meiofaunal taxa.
Line 257. Please, use “taxonomic composition” instead of “community structure”.
Line 271. What Authors mean with “frequently”?
Line 281. Please, indicate “µg C” only once.
Line 286. At all sites? And maybe is “going deeper”.
Line 288. Never heard about “Nematode structural indices”. It has no sense. MI indicates life strategies and not functional diversity.
Line 323. “Faunal descriptors (i.e., meiofauna abundance, community composition, number of major taxa,”. The term “descriptor” is not common, above all for community composition (which should be “taxonomic composition”).
Line 327. Personally, I do not like the term “nematofauna” because it could comprise also macrobenthic nematodes. However, I recognize that it is matter of personal opinion.

Discussion
Line 339. I think that hydrothermal vents are not biotopes.
Line 358. Please, replace “density” with “abundance”.
Line 359. Please, replace “abundance” with “amount” or “quantitative”.
Line 366. Only in Mediterranean Sea?
Line 370-371. Both in vents and no-vents systems?
Line 384. Again, the differences are not significant.
Line 405. The term “permanent” is not appropriate, since no temporal analyses have been conducted.
Line 409. Are only living forams here?
Line 404. What about other meiofaunal taxa? There is any exclusive taxon at each site?
Line 438. Please, do not use “populations”.
Line 503. Where correlation analysis has been carried out?
Line 543. Replace “community with “assemblage”.

Figures and Tables
Figure 1. What are “white” and “yellow” vents?
Figure 5 and 6. To my knowledge chaetognata are planktonic. Copepoda and nauplii should be merged together, being the same taxon.
Table 2. What does it means “All replicates are reported”?. I guess the avg is reported? Or cumulative indexes?
Table 3. I guess Var% is not expressed in 100. Otherwise the values are too low.

Reviewer 3 ·

Basic reporting

This manuscript reported the meiofauna, especially marine nematodes communities around a hydrothermal vent in the shallow water. Generally, the materials and methods are suitable. The results are reliable and the manuscript was well organized and written.

Experimental design

There are totally four sites selected for meiofaunal and nematodes samples collecting with different distances to the vent. This experimental design is suitable to reflect the effect of hydrothermal vent on meiofaunal and nematodes communities. The laboratory processing and data analysis are both OK.

Validity of the findings

The findings of this study are reliable based on the data presentation.

Additional comments

I have two suggestions for the improvement of the manuscript.
(1) Figure 1 is not clear and not friendly for readers.
(2) The methods to collect meiofaunal samples are not clear. By remote sediment corers or by diving or by submersibles/ROV? Please state it clearly.

---

## Round 0.2 · accepted · Accept

Please pay close attention to the minor comments offered by the reviewers, these will benefit your manuscript.

·

Basic reporting

After the three reviewers suggestions the MS greatly improved. Now the text is clearer and fluent. I observed some mistyping but I suggest to the editor not to resubmit the manuscript by the authors but to have them corrected during the proofs revision by the authors themselves.

These are the mistyping:
Line 50: Extra space at the beggining of the sentence.
Line 74: the same as for line 50.
Line 195: please change Log(x+1)[(x+1) as subscript] with log(x+1).

I accept all the corrections made by the authors that diligently followed my and other reviewers suggestions.

Experimental design

I already stated in the first review that the experimental design is good for the questions the authors wanted to answer.

Validity of the findings

As stated in the first review, the findings are valid and relevant.

Additional comments

I congratulate the authors for the good work made.

Reviewer 2 ·

Basic reporting

The manuscript has been improved, considering almost all the suggested comments.

Experimental design

The requested details have been added.

Validity of the findings

The validity has been recognized at the first stage of the submission.

Additional comments

Overall, the manuscript has been improved, considering almost all the suggested comments.

In some cases, Authors did not accept the suggestions (e.g., keeping separated different life stages even belonging to the same taxa or calculating the diversity indexes. I am still convinced that these are errors, from an ecological point of view, but I recognize that it is matter of personal opinion.
I think also that at the end the Scientific Community will evaluate the paper, citing it or not.

Reviewer 3 ·

Basic reporting

This manuscript has been improved significantly after revision.

Experimental design

no comment.

Validity of the findings

no comment.

Additional comments

This manuscript has been improved significantly after revision. It should be acceptable for publication.